# Prior Placement of Male Urethral Slings Can Increase the Need for Revision of Artificial Urinary Sphincters

**DOI:** 10.3390/jcm10245842

**Published:** 2021-12-13

**Authors:** Emily M. Yura, Christopher J. Staniorski, Jason E. Cohen, Liqi Chen, Ashima Singal, Francisco E. Martins, Matthias D. Hofer

**Affiliations:** 1Department of Urology, Northwestern University Feinberg School of Medicine, Chicago, IL 60611, USA; emmy.yura@gmail.com (E.M.Y.); staniorskicj@upmc.edu (C.J.S.); jason.cohen@northwestern.edu (J.E.C.); ashima.singal@gmail.com (A.S.); 2Department of Preventative Medicine Biostatistics Collaboration Center, Northwestern University Feinberg School of Medicine, Chicago, IL 60611, USA; liqi.chen@northwestern.edu; 3Department of Urology, University of Lisbon, School of Medicine, 1649-028 Lisbon, Portugal; faemartins@gmail.com; 4Urology San Antonio, San Antonio, TX 78258, USA

**Keywords:** male incontinence, male urethral sling, artificial urinary sphincter, urethral

## Abstract

Background: Recurrent stress urinary incontinence (SUI) following male sling can be managed surgically with artificial urinary sphincter (AUS) insertion. Prior small, single-center retrospective studies have not demonstrated an association between having failed a sling procedure and worse AUS outcomes. The aim of this study was to compare outcomes of primary AUS placement in men who had or had not undergone a previous sling procedure. Methods: A retrospective review of all AUS devices implanted at a single academic center during 2000–2018 was performed. After excluding secondary AUS placements, revision and explant procedures, 135 patients were included in this study, of which 19 (14.1%) patients had undergone prior sling procedures. Results: There was no significant difference in demographic characteristics between patients undergoing AUS placement with or without a prior sling procedure. Average follow up time was 28.0 months. Prior sling was associated with shorter overall device survival, with an increased likelihood of requiring revision or replacement of the device (OR 4.2 (1.3–13.2), *p* = 0.015) as well as reoperation for any reason (OR 3.5 (1.2–9.9), *p* = 0.019). While not statistically significant, patients with a prior sling were more likely to note persistent incontinence at most recent follow up (68.8% vs. 42.7%, *p* = 0.10). Conclusions: Having undergone a prior sling procedure is associated with shorter device survival and need for revision or replacement surgery. When considering patients for sling procedures, patients should be counseled regarding the potential for worse AUS outcomes should they require additional anti-incontinence procedures following a failed sling.

## 1. Introduction

Male stress urinary incontinence (SUI) is associated with decreased quality of life. As a perpetual problem, it can cause considerable distress to patients. While radical prostatectomy accounts for the majority of men suffering with SUI, additional causes for male SUI include neurologic conditions, prior pelvic radiation, and prior transurethral or urethral surgeries. Conservative treatment options for male SUI include behavioral therapy, pelvic floor exercises, or pharmacotherapy. However, an estimated 5–6% of patients with post-prostatectomy SUI will need to proceed to surgical treatment of incontinence [1,2].

In men who have persistent, severe SUI refractory to conservative treatment, AUS placement remains the gold standard treatment, with long-term success in achieving acceptable continence (less than or equal to one pad per day) ranging from 59–90% [3]. Despite excellent success in the treatment of SUI, however, AUS surgery is associated with complications, with a reported 3.3–27.8% of cases experiencing infection/erosion, 2.0–13.8% with mechanical failure, and 1.9–28.6% with urethral atrophy [4]. Amongst those with an AUS, 14.8–44.8% require reintervention such as explanation for erosion or infection and/or revision for recurrent or persistent leakage, with an average of 1.5 reoperations required over time [4].

As an alternative to AUS, several male sling devices have been developed. The success rate of slings to achieve acceptable continence varies widely by report ranging from 8.3–92% with a similarly wide range of reported complications requiring explanation (0.6–35%) [5]. However, despite advances in sling technology, there remains the pervasive issue of declining efficacy over time, possibly due to urethral atrophy secondary to long-standing compression of the corpus spongiosum and decreased urethral blood supply. Adjustable slings, which supposedly minimize pressure impact on the urethra, have been developed but are not available in most countries.

Despite more modest success rates and less robust clinical investigations, men may opt for sling placement over AUS to avoid the complexity and complications associated with mechanical devices. In one study, 25% of patients with high grade SUI who had been advised by their surgeon to undergo AUS proceeded with sling placement against their surgeon’s recommendations [6]. Perhaps reflecting a patient preference towards slings, the proportion of sling procedures being performed for treatment of male SUI appears to have increased relative to AUS procedures amongst certifying and re-certifying urologists [7]. 

An often-accepted tenet amongst urologists treating men with SUI is that a sling may be performed without compromising subsequent AUS procedures, and in one study, 13% of patients undergoing sling procedure ultimately underwent AUS procedure for persistent incontinence [1]. However, the evidence regarding AUS success following a failed sling is scarce, limited to a few, small, single-center, retrospective cohort studies with somewhat conflicting evidence regarding the outcomes [8,9]. The aim of this study was to investigate the outcomes of AUS procedures at our institution in patients with and without a history of male sling placement for SUI in order to optimize patient selection for male SUI surgery.

## 2. Materials and Methods

### 2.1. Patient Selection

The study was conducted in accordance with the Declaration of Helsinki, and the protocol was approved by the Institutional Review Board of Northwestern University. We retrospectively reviewed the records of all patients who underwent placement of an artificial urinary sphincter at a single academic institution between 2000 and 2018. All AUS surgeries performed at our institution used the AMS800 (Boston Scientific, Marlborough, MA, USA) device. We utilized the institution’s Electronic Data Warehouse to screen and identify patients using CPT codes for AUS surgery. For the purpose of this study, we included primary AUS procedures for the management of SUI in male patients over the age of 18. We excluded procedures which were for revision or replacement of AUS and AUS cuff placement to the bladder neck. The indication for AUS placement in all patients was bothersome stress urinary incontinence.

### 2.2. Data Collection and Outcome Definition

All patient charts were reviewed for the following information at the time of surgery: demographic data, etiology of incontinence, past medical, surgical and social history, medication list, surgical approach, operative time, and AUS cuff size and location. In patients with a prior male sling, management of the sling material was documented. Serum testosterone levels were extracted by query of the medical record with values drawn within two years of the procedure included in analysis; if multiple values were found, the value closest to the date of surgery was used for analysis. Hypogonadism was defined as a total testosterone level <300 ng/dL on the specified result.

Patient follow-up was analyzed at an early timepoint (within 12 months following the procedure) as well as at the most recent visit (>12 months). Continence status was included only for those with a formal follow-up appointment and clear documentation of status or pad use. Continence was defined as the use of 0–1 pads per day (safety pad use). AUS re-operations were divided into those that were performed for erosion or infection of the AUS and those that were for revision (e.g., due to persistent incontinence). While each etiology for reoperation was analyzed separately, we also analyzed both entities together as revision for any cause.

### 2.3. Statistical Analysis

Statistical analysis was performed with SPSS 20 for Mac (IBM, Armonk, NY, USA). Statistical tests performed included two-sample *t*-test (continuous characteristics), Chi-squared tests (categorical characteristics), and univariate and multivariate logistic regression analysis. Additionally, we used Kaplan-Meier curves to assess association between prior sling and time to erosion and infection, device revision, and overall device survival.

## 3. Results

We included 135 primary AUS placements of which 19/135 (14.1%) patients had a prior sling while 116/135 (85.9%) patients had not. Patient characteristics are listed in Table 1 demonstrating that there were no significant differences among groups including preoperative pad use, co-morbidities or known risk factors for AUS erosions such as pelvic radiation, hypogonadal state, or prior urethral surgery.

In our study a variety of slings were used: AdVance (2/19 (10.5%)), Virtue (3/19, (15.8%)), InVance (2/19, (10.5%)), Stamey-type sling (11/19 (57.9%)), and one unknown sling type (1/19 (5.3%)). Sling surgery was performed a mean of 65.6 months prior to AUS insertion (range 6.5–223). In 9/19 cases (47.4%) the sling was explanted at time of AUS placement and in the remaining cases it was left in place. In the remaining cases the sling was left in place which was performed when removal of the sling would have posed a significant risk for urethral injury. Notably, there was no association between device failure and absence/presence of a sling after AUS placement.

Operative time for AUS placement was significantly longer in patients that had a prior sling compared to those that did not (140.88 vs. 109.51 min, *p* = 0.008) while other operative parameters did not differ (Table 2).

Postoperative continence outcomes are summarized in Table 2. While there was no difference in continence rates within 12 months of AUS placement, we found that at overall follow up, patients with a prior sling had a significantly lower continence rate (4/15 (26.7% vs. 48/83 (67.8%), *p* = 0.026) at a mean follow-up of 28.06 months (2–176 months). There was no statistically significant difference in follow up duration between the two groups.

As demonstrated in Table 3, we did not find a statistically significant difference in the need for AUS explanation for infection or erosion. However, patients that had a prior sling required a revision of their AUS significantly more frequently (7/19 (36.8%) vs. 14/116 (12.1%), *p* = 0.006). This was still significantly different when combining all reoperations encountered (10/19 (52.6%) vs. 29/116 (25%), *p* = 0.014).

On univariate logistic regression analysis (Table 4), we analyzed the effect of prior sling, age, presence of co-morbidities that may influence the vascular supply of the urethra such as diabetes and peripheral vascular disease, and known risk factors for sling erosion such as pelvic radiation, presence of a 3.5 cm cuff, or prior urethroplasty. We did not include hypogonadal status as testosterone levels were only available for 31 patients (23%). While there were no significant risk factors for AUS erosion or infection, only presence of a prior sling was significantly associated with revision or any reoperation conferring an odds ratio of 4.2 and 3.5, respectively (*p* = 0.015 and 0.019). In a multivariable regression analysis including the same parameters, prior sling status was also independently associated with the risk for revision (*p* = 0.010) or reoperation for any cause (*p* = 0.017) (Appendix A).

We also performed a Kaplan-Meier analysis (Figure 1) to analyze postoperative AUS survival over time. Presence of prior sling was significantly associated with decreased AUS survival in general (need for reoperation for any cause, log rank *p* = 0.042) and with decreased survival until a revision was necessary (log rank *p* = 0.02).

## 4. Discussion

Male sling procedures offer an appealing solution to men suffering with SUI primarily because, as a passive device, it avoids the need for actively operating a mechanical pump as with an AUS. However, while slings provide durable continence in carefully selected patients [10,11], some men may opt for sling placement despite being imperfect candidates to avoid the shortcomings of an AUS [6]. This decision is justified by the concept that a AUS can be placed if a sling fails, as a sling will not compromise subsequent AUS outcomes. However, this notion is based on less than a handful of reports. In the current study, we found that prior sling placement is associated with decreased continence rates and an increased AUS revision rate conferring a more than four-fold risk.

Our results contradict aforementioned prior studies which have suggested non-inferior AUS outcomes in patients with prior slings. For example, Lentz et al. found similar AUS device survival amongst 29 men with and 136 men without a prior sling at a mean follow up of less than 2 years [8]. Based on our results, the AUS survival curves start to differ at about 2 years post-AUS implantation and therefore the authors likely did not capture all AUS failures in their patients that had a prior sling. In addition, the follow-up for patients with prior sling was significantly lower than for those without (20.7 vs. 37.2 months, *p* < 0.005) which potentially skews the relative incidence of device complications further. A recent study by Ziegelmann et al. is equally limited and skewed in follow-up with 1.8 years for patients with prior slings (compared to 3.2 years for those without, *p* = 0.008), missing later AUS failures in patients after sling [9]. The authors report a 15% decreased AUS survival in patients with prior sling and while this difference was not statistically significant at time of analysis, our data suggest that a few months of further follow-up may have been sufficient to find statistical significance. Additionally, neither study compared long-term continence outcomes, but given the considerably short follow-up, it is unclear whether this would have been captured. Giammo et al. analyzed the efficacy of the ATOMS system and found that prior incontinence surgery (the vast majority of cases received a ProACT device which provides lateral compression to the urethra via two inflated balloons similar to compression by a sling) was significantly associated with persistent incontinence after ATOMS placement [12].

We believe that chronic urethral compression by a sling decreases urethral blood flow and results in urethral atrophy over time, subsequently compromising both sling and AUS outcomes. Urethral atrophy distal to a sling is demonstrated in Figure 2. Urethral atrophy is associated with a thinned corpus spongiosum, and this confers decreased compressive pressure after sling placement, facilitating recurrence of urinary leakage over time. Likewise, a thin corpus spongiosum due to urethral atrophy makes sufficient coaptation by AUS cuff more difficult to achieve. Given that we did not find a difference in erosion rates it appears that the urethral atrophy and spongiosal thinning is not critical enough to facilitate erosion. Therefore, it should be emphasized that we do not question whether an AUS can or should be placed after a failed sling surgery, but we feel that long-term continence rates are less successful and hence the need for device revision increases. Indeed, we discovered that men with a prior sling experienced shorter device survival, with >4-fold increased odds of revision as well as 3.5-fold increased odds of reoperation for any reason. Therefore, we believe that patients who are likely to require an AUS in the future should be counseled that primary AUS placement may be associated with longer device survival and decreased likelihood of needing AUS revision compared to AUS placement following male sling procedure. In general, we reserve slings for patients that use three or less pads per day and have not received radiotherapy and are unlikely to receive it in the future (stable undetectable PSA levels).

Unlike other studies on the subject, our sling patients have been treated with a variety of sling procedures including Virtue, AdVance, InVance, and Stamey slings. The Stamey sling consists of placement of a hammock of GoreTex vascular graft bolsters placed suburethrally and external to the bulbospongiousus muscle. The hammock strings are passed retropubically through the rectus muscle and fascia and then tightened. As with all slings, continence is achieved by urethral compression.

Limitations of our study include its retrospective design which among the inert biases of such a study prevents us from analyzing validated questionnaires as this data, while collected, was not included in the electronic health record. The variety of slings utilized, and management of sling material confers heterogeneity to the sling cohort; however, we feel that it is not the type of sling inserted but rather the principle of chronic compression of the urethral blood supply which contributes to urethral atrophy and the ultimate risk of AUS failure. We feel that we were able to collect a considerable number of patients with prior sling although this number still limits overall conclusions. With regards to follow up, the range in follow-up duration may introduce bias as some complications that take longer to occur than others (e.g., sphincter erosion) may not have all been captured. We also did not routinely measure 24-h pad weights to assess postoperative continence outcomes, we instead documented number of pads per day as a surrogate, which, although less accurate than pad weight tests, is routinely utilized in incontinence studies. Additionally, as restoration of quality of life is a predominant objective of any incontinence surgery, recurrent urinary incontinence may not reflect patient bother if overall improved compared to baseline.

## 5. Conclusions

Patients with a prior sling procedure were more likely to experience recurrent incontinence after AUS placement and had a four-fold risk to require revision of their AUS. This underlines the importance of patient selection for sling placement and advocates for appropriate counseling of patients to consider primary AUS placement.

## Figures and Tables

**Figure 1 jcm-10-05842-f001:**
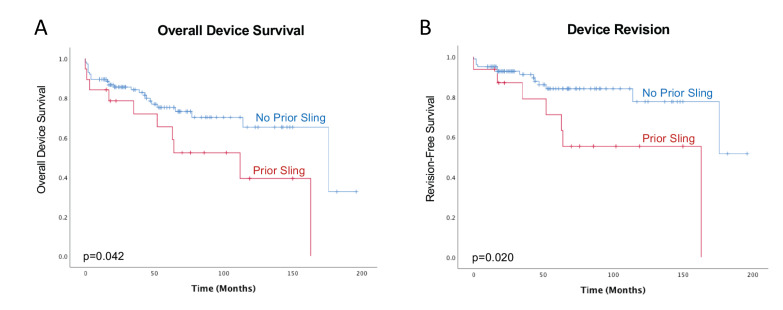
Kaplan-Meier analyses: Presence of a prior sling significantly decreases AUS survival postoperatively, both overall (need for reoperation for any cause, log rank *p* = 0.042, panel (**A**)) and until a revision was needed for insufficient continence (log rank *p* = 0.02, Panel (**B**)).

**Figure 2 jcm-10-05842-f002:**
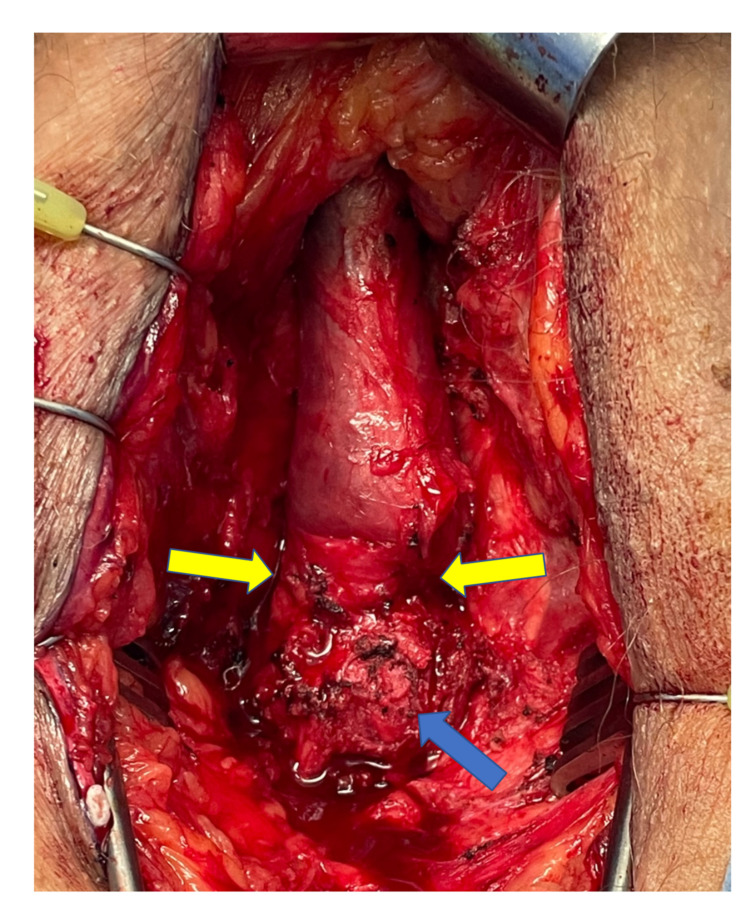
Urethral atrophy after sling placement: The sling (AdVance) is located at the bulbar urethra (blue arrow). Note the caliber change of the urethra distal to it, indicated by yellow arrows.

**Table 1 jcm-10-05842-t001:** Patient Characteristics.

	No Prior Sling	Prior Sling	*p*-Value
Age (range)	66.0 (23–83)	64.5 (20–84)	0.582 *
BMI (range)	28.8 (19–46)	29.3 (22–37)	0.701 *
Smoking History	102/108 (94.4%)	14/17 (82.4%)	0.730 ^§^
Diabetes	17/116 (14.7%)	3/19 (15.8%)	0.897 ^§^
PVD	4/116 (3.4%)	0/19 (0%)	0.411 ^§^
Prostate Ca	104/116 (89.7%)	17/19 (89.5%)	0.981 ^§^
Preop pads/day (range)	3.56 (1–12)	4.69 (1–10)	0.079 *
Neurogenic Bladder	5/116 (4.3%)	0/19 (0%)	0.356 ^§^
RRP	96/116 (82.8%)	17/19 (89.5%)	0.463 ^§^
XRT	39/116 (33.6%)	4/19 (21.0%)	0.276 ^§^
Urethral Surgery	16/116 (13.8%)	1/19 (5.3%)	0.299 ^§^
BPH Procedure	13/116 (11.2%)	1/19 (5.3%)	0.431 ^§^
ADT	16/116 (13.8%)	2/19 (10.5%)	0.698 ^§^
Hypogonadal	18/26 (69.2%)	3/5 (60.0%)	0.686 ^§^

* Student’s *t*-test. ^§^ Chi-squared test. BMI = body mass index, PVD = peripheral vascular disease, RRP = radical retropubic prostatectomy, XRT = radiation, BPH = benign prostatic hyperplasia, ADT = androgen deprivation therapy.

**Table 2 jcm-10-05842-t002:** Intraoperative Characteristics and Postoperative Outcomes.

	No Prior Sling	Prior Sling	*p*-Value
Operative Information			
Mean OR Time (min) (range)	109.5 (57–265)	140.9 (62–366)	0.008 *
3.5 cm Cuff	21/116 (18.1%)	5/19 (26.3%)	0.400 ^§^
Tandem Cuff	5/116 (4.3%)	2/19 (10.5%)	0.257 ^§^
Transcorporal Cuff	11/116 (9.5%)	1/19 (5.3%)	0.549 ^§^
Concurrent IPP	6/116 (5.2%)	3/16 (2.6%)	0.085 ^§^
Early follow up (<12 months)			
Mean follow up (months) (range)	4.22 (2–10)	4.14 (2–10)	0.891 *
Continent	59/72 (81.9%)	10/14 (71.4%)	0.366 ^§^
Mean pads per day (range)	1.15 (0–3)	0.86 (0–1)	0.235 *
Overall follow up			
Mean follow up (months) (range)	30.28 (2–175)	45.77 (3–163)	0.132 *
Continent	48/83 (57.8%)	4/15 (26.7%)	0.026 ^§^
Mean pads per day	1.52 (0–6)	2 (0–4)	0.409 *

* Student’s *t*-test. ^§^ Chi-squared test.

**Table 3 jcm-10-05842-t003:** Need for Reoperation.

	No Prior Sling	Prior Sling	*p*-Value
Mean f/u (months) (range)	25.8 (0–176)	41.7 (0–163)	0.089 *
Erosion or Infection	15/116 (12.9%)	3/19 (15.8%)	0.734 ^§^
Revision	14/116 (12.1%)	7/19 (36.8%)	0.006 ^§^
Any Repeat Operation	29/116 (25.0%)	10/19 (52.6%)	0.014 ^§^

* Student’s *t*-test. ^§^ Chi-squared test.

**Table 4 jcm-10-05842-t004:** Univariate Analysis.

Erosion/Infection	*p*-Value	OR	95% C.I.	
			Lower	Upper
Age > 65 years	0.839	1.127	0.355	3.578
Diabetes	0.653	1.375	0.344	5.498
PVD	0.614	1.856	0.168	20.448
Radiation	0.21	1.982	0.68	5.775
Urethroplasty	0.767	0.783	0.155	3.953
Prior Sling	0.648	1.385	0.341	5.627
3.5 cm cuff	0.685	1.292	0.375	4.46
**Revision**	***p*-Value**	**OR**	**95% C.I.**	
			**Lower**	**Upper**
Age > 65 years	0.629	1.305	0.443	3.852
Diabetes	0.18	2.298	0.68	7.758
PVD	0.999	0	0	
Radiation	0.66	0.77	0.24	2.467
Urethroplasty	0.693	1.338	0.316	5.664
Prior Sling	0.015	4.183	1.328	13.181
3.5 cm cuff	0.113	2.422	0.81	7.244
**Any Reoperation**	***p*-Value**	**OR**	**95% C.I.**	
			**Lower**	**Upper**
Age > 65 years	0.585	1.277	0.531	3.067
Diabetes	0.165	2.1	0.738	5.979
PVD	0.832	0.774	0.073	8.262
Radiation	0.514	1.336	0.56	3.191
Urethroplasty	0.959	1.031	0.318	3.345
Prior Sling	0.019	3.496	1.229	9.947
3.5 cm cuff	0.111	2.131	0.841	5.401

PVD = peripheral vascular disease.

## Data Availability

The data is available upon request from the authors.

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
