# Peer review of "Prior Placement of Male Urethral Slings Can Increase the Need for Revision of Artificial Urinary Sphincters"

_jcm, 2021, doi:10.3390/jcm10245842_

Round 1

Reviewer 1 Report

Dear Authors,

I read with interest your paper entitled “Prior Placement of Male Urethral Slings Can Increase the Need for Revision of Artificial Urinary Sphincters”.

Notwithstanding the article is interesting, it is not devoid from minor limitations.

Please find here a list of suggested modifications:

  • Unlike sling, it was not specified which AUS models were placed during the study, please specify. If this information is not available address it in the study limits.
  • It is unclear whether all patients undergoing AUS placement after slingi were candidates for this procedure solely because of persistence of urinary incontinence, please specify. In addition, the authors state that in 9/19 cases (47.4%) the sling was explanted at time of AUS placement. How were the remaining 10 patients managed?
  • Please correct table 1, there are some grammatical errors
  • Please correct line 135: “ncountered (10/19 (52.6%) vs. 29/116 (25%), p=0.014)”.

Please consider to cite the following pertinent papers:

  • Minerva Urol Nefrol.2019 Nov 4. doi: 10.23736/S0393-2249.19.03457-X. Implant of ATOMS® system for the treatment of postoperative male stress urinary incontinence: an Italian multicentric study. Giammò A1, Ammirati E2, Tullio A3, Morgia G4, Sandri S5, Introini C6, Canepa G6, Timossi L7, Rossi C8, Mozzi C8, Carone R
  • Minerva Urologica e Nefrologica 2020 October;72(5):555-62. Conservative management of urinary incontinence following robot-assisted radical prostatectomy. Michele MARCHIONI *, Giulia PRIMICERI, Pietro CASTELLAN, Luigi SCHIPS, Guglielmo MANTICA, Christopher CHAPPLE, Rocco PAPALIA, Francesco PORPIGLIA, Roberto M. SCARPA, Francesco ESPERTO

Author Response

Thank you for reviewing our paper and your valuable comments. I am addressing your comments point-by-point below:

“Notwithstanding the article is interesting, it is not devoid from minor limitations.”

Thank you for your kind comment and we are addressing the limitations you are raising below.

“Unlike sling, it was not specified which AUS models were placed during the study, please specify. If this information is not available address it in the study limits.”

The only model that was placed was the AMS800 (Boston Scientific, Marlborough, MA, USA). We have added this to the manuscript. We added following sentence in the Methods section:

“All AUS surgeries performed at our institution used the AMS800 (Boston Scientific Marlborough, MA, USA) device.”

“It is unclear whether all patients undergoing AUS placement after sling were candidates for this procedure solely because of persistence of urinary incontinence, please specify.

We have added the following sentence to the Methods section for clarification:

“The indication for AUS placement in all patients was bothersome stress urinary incontinence.”

“In addition, the authors state that in 9/19 cases (47.4%) the sling was explanted at time of AUS placement. How were the remaining 10 patients managed?”

In the remaining cases the sling was left in place which was done when removal of the sling would have posed a significant risk for urethral injury. Notably, there was no association between device failure and absence/presence of a sling after AUS placement.

This sentence was verbatim included in the Results section.

“Please correct table 1, there are some grammatical errors”

The errors have been corrected.

“Please correct line 135: “ncountered (10/19 (52.6%) vs. 29/116 (25%), p=0.014)”.

 This has been corrected.

“Please consider to cite the following pertinent papers:

Minerva Urol Nefrol.2019 Nov 4. doi: 10.23736/S0393-2249.19.03457-X. Implant of ATOMS® system for the treatment of postoperative male stress urinary incontinence: an Italian multicentric study. Giammò A1, Ammirati E2, Tullio A3, Morgia G4, Sandri S5, Introini C6, Canepa G6, Timossi L7, Rossi C8, Mozzi C8, Carone R”

We have included this reference and would like to thank the Reviewer for suggesting it.

Reviewer 2 Report

The authors used a retrospective method to compare slings and AUS in the surgical treatment of postprostatectomy urinary incontinence. They confirmed that the highest degree of urinary incontinence is not optimal for primary sling implantation because it worsens the results for secondary AUS implantation. The limiting factor of the study is the low number of patients in the sling group, and therefore the conclusions should be taken with some caution for bias.

Please correct number of patients Line 19/109:

Line 19 - "20 (14.8%) patients had undergone prior sling procedures"

Line 106 - "of which 19/135 (14.1%) patients had a prior sling"

Line 100 - Which statistical values were used - mean/SD or median/percentile? The median and percentile are more suitable for nonparametric data distribution in this study.

Line 129 - Which statistical values were used - mean or median in the Table 1?

Line 111 - What was the initial degree of postprostatectomy urinary incontinence before the primary implantation of the sling/AUS? Why was not the primary AUS but the sling implanted at the highest degree of urinary incontinence?

Line 129, 132 - Explain all the abbreviations in tables to make it clear to all readers

Line 235 - Add to limitation the low number of patients in the sling group vs bias of study

Line 250 - It is useful for the reader to write a degree of postprostatectomy urinary incontinence that is suitable for primary sling / AUS implantation because secondary post-sling AUS implantation often fails.

Author Response

Thank you for reviewing our paper and your valuable comments. I am addressing your comments point-by-point below:

The authors used a retrospective method to compare slings and AUS in the surgical treatment of postprostatectomy urinary incontinence. They confirmed that the highest degree of urinary incontinence is not optimal for primary sling implantation because it worsens the results for secondary AUS implantation. The limiting factor of the study is the low number of patients in the sling group, and therefore the conclusions should be taken with some caution for bias.

Please correct number of patients Line 19/109:

Line 19 - "20 (14.8%) patients had undergone prior sling procedures"

Line 106 - "of which 19/135 (14.1%) patients had a prior sling"

We have corrected this and it is now stated in the Abstract:

“After excluding secondary AUS placements, revision and explant procedures, 135 patients were included in this study, of which 19 (14.1%) patients had undergone prior sling procedures.”

“Line 100 - Which statistical values were used - mean/SD or median/percentile? The median and percentile are more suitable for nonparametric data distribution in this study.

We used the mean value in this study. We agree that median and percentile would have been a good alternative, but our statistician felt that the distribution would justify using the mean value.

“Line 129 - Which statistical values were used - mean or median in the Table 1?”

We used the mean for age and BMI and t-test for continuous and chi-square for categorical variables.

“Line 111 - What was the initial degree of postprostatectomy urinary incontinence before the primary implantation of the sling/AUS? Why was not the primary AUS but the sling implanted at the highest degree of urinary incontinence?”

The degree of incontinence varied, Stamey slings were implanted primarily in patients using 2 to 10 pads per day (5.6 in average). For the remaining slings, 1 to 4 pads per day (3.1 in average) was used. In the earlier years of the study, Stamey sling placement was seen as a valid alternative to primary AUS placement.

“Line 129, 132 - Explain all the abbreviations in tables to make it clear to all readers”

We have inserted the following in Table 1:

“BMI=body mass index, PVD=peripheral vascular disease, RRP=radical retropubic prostatectomy, XRT=radiation, BPH=benign prostatic hyperplasia, ADT=androgen deprivation therapy.”

And in Table 4:

“PVD=peripheral vascular disease“

Reviewer 3 Report

#1: The authors mentioned the following sentences in the abstract: "Prior sling 21 was associated with shorter overall device survival, with an increased likelihood of requiring revision or replacement of the device (OR 5.23 (1.8-15.2), p=0.003) as well as reoperation for any reason (OR 3.8 (1.4-10.1), p=0.008)". However, I could not find out any data presentation in the result section or Tables which support aforementioned sentences.

#2: The authors conducted logistic regression analysis. Please describe it in the method section.

#3: The authors conducted t-test for comparison of continuous data; however, I think Wilcoxon rank sum test is suitable for it.

#4: Please show OR and 95%CI in the multivariate analysis.

#5: Please add abbreviations in the foot of Tables.

#6: After addressing my queries I suggested (#1 to #5), I will try to review your manuscript again.

Author Response

Thank you for reviewing our paper and your valuable comments. I am addressing your comments point-by-point below:

“#1: The authors mentioned the following sentences in the abstract: "Prior sling 21 was associated with shorter overall device survival, with an increased likelihood of requiring revision or replacement of the device (OR 5.23 (1.8-15.2), p=0.003) as well as reoperation for any reason (OR 3.8 (1.4-10.1), p=0.008)". However, I could not find out any data presentation in the result section or Tables which support aforementioned sentences”.

We have corrected this data discrepancy between abstract and tables and it is now stated in the Abstract:

“Prior sling was associated with shorter overall device survival, with an increased likelihood of requiring revision or replacement of the device (OR 4.2 (1.3-13.2), p=0.015) as well as reoperation for any reason (OR 3.5 (1.2-9.9), p=0.019)”

“#2: The authors conducted logistic regression analysis. Please describe it in the method section.”

We have revised this, and it now reads”

“Statistical analysis was performed with SPSS 20 for Mac (IBM, Armonk, NY).  Statistical tests performed included two-sample t-test (continuous characteristics), chi-squared tests (categorical characteristics), and univariate and multivariate logistic regression analysis”

“#3: The authors conducted t-test for comparison of continuous data; however, I think Wilcoxon rank sum test is suitable for it.”

We agree that a Wilcoxon rank sum test would have been suitable but our statistician felt that a t-test was appropriate.

“#4: Please show OR and 95%CI in the multivariate analysis.”

We have updated the Table as requested.

“#5: Please add abbreviations in the foot of Tables.”

The abbreviations have been added. We have inserted the following in Table 1:

“BMI=body mass index, PVD=peripheral vascular disease, RRP=radical retropubic prostatectomy, XRT=radiation, BPH=benign prostatic hyperplasia, ADT=androgen deprivation therapy.”

And in Table 4:

“PVD=peripheral vascular disease“

“#6: After addressing my queries I suggested (#1 to #5), I will try to review your manuscript again.”

Thank you -we feel we have addressed all your comments and hope that will agree.

Round 2

Reviewer 3 Report

None.